# Systematic Review of Actinomycetes in the Baijiu Fermentation Microbiome

**DOI:** 10.3390/foods11223551

**Published:** 2022-11-08

**Authors:** Cong Chen, Haiquan Yang, Jie Liu, Huibo Luo, Wei Zou

**Affiliations:** 1College of Bioengineering, Sichuan University of Science & Engineering, Yibin 644005, China; 2The Key Laboratory of Industrial Biotechnology, Ministry of Education, School of Biotechnology, Jiangnan University, Wuxi 214122, China; 3Anhui Linshui Liquor Co., Ltd., Lu’an 237471, China

**Keywords:** actinomycete, Baijiu, flavor compound, interspecies interactions, pit mud

## Abstract

Actinomycetes (a group of filamentous bacteria) are the dominant microbial order in the Daqu (DQ) fermentation starter and in the pit mud (PM) of the Baijiu fermentation microbiome. Actinomycetes produce many of the key enzymes and flavor components, and supply important precursors, which have a major influence on its characteristic aroma components, to other microorganisms during fermentation. This paper reviews the current progress on actinomycete research related to Baijiu fermentation, including the isolation and identification, distribution, interspecies interactions, systems biology, and main metabolites. The main metabolites and applications of the actinomycetes during Baijiu fermentation are also discussed.

## 1. Introduction

Baijiu is one of the six major distilled liquors in the world and has been produced for centuries in China [1,2]. Baijiu is colorless, clear, and has a unique flavor, which mainly arises from the co-fermentation of microorganisms naturally inoculated. This method results in a very wide range of fermenting organisms, from both the Daqu (DQ) saccharification starter, the pit mud (PM), and the fermented grains (FGs) [3,4,5]. Depending on its flavor characteristics caused by the different production processes, raw materials, and microbial communities, Baijiu can be classified into the four basic Baijiu aroma types—jiangxiangxing (JXXB), nongxiangxing (NXXB), qingxiangxing (QXXB), andmixiangxing (MXXB)—and eight Baijiu aroma types derived from the above four aroma types [1]. Baijiu production consists of four successive stages: (i) saccharification of the grain starch using DQ or Xiaoqu (XQ) as the starter culture; (ii) open solid-state fermentation; (iii) distillation to produce the final liquor product; and (iv) storage aging. The open saccharification fermentation results in a highly complex microbiome, with many interactions between species [6,7]. This huge microbial diversity is responsible for the unique flavor of Baijiu, which is distinct from those of other distilled liquors, such as whisky, vodka, and tequila, which are made by submerged fermentation [8]. In essence, the diversity and stability of microbial communities associated with high acid tolerance are directly influenced by factors of the natural environmental conditions (geographical limitations, temperature, humidity, pH, and climate), raw materials (sorghum or a mixture of wheat, barley, corn, rice, and sorghum), and complex production processes [9,10]. The high complexity of the naturally selected microbiome has the potential to produce distinct flavors containing different trace components and alcohol contents (35–60%), caused by their underlying metabolism and interactions.

Actinomycetes in Baijiu fermentation derive from the DQ or XQ starter, the PM, and the fermenting grains (FGs) [11,12]. They have diverse metabolic activities, including the hydrolysis of starch, cellulose, protein, and pectin in the raw material grains, and they produce various secondary metabolites, including flavor esters (ethyl caproate, ethyl butyrate, and ethyl lactate), and other flavor compounds (3-hydroxyl-2-butanone, 2,3-butanediol, and pyrazine), polypeptides, and amino acids [13]. In DQ-initiated fermentation, actinomycetes produce a large amount and diversity of antibiotics, such as heptaene macrolide antibiotics and streptomycin, which can inhibit the growth of fungi and pathogenic bacteria [14,15]; in particular, *Streptomyces* spp., which produces geosmin that can spoil the flavor of the Baijiu in the early stages of saccharification and fermentation [16,17]. Several reports have focused on the application of actinomycete topromotecaproic acid bacteria to produce ethyl caproate, the main contributor to the flavor of nongxiangxing Baijiu [18,19]. In addition, actinomycetes reportedly help to maintain the stability of the PM and accelerate the maturation of new PM [20,21]. This review focuses on the recent research progress on actinomycetes in the Baijiu microbiome, including the distribution and abundance and species of actinomycetes in DQ and PM, ‘omics studies, and interactions between actinomycetes. The functions of actinomycetes isolated from Baijiu fermentations are also discussed.

## 2. Isolation and Identification of Actinomycetes from Different Stages of Baijiu Production

In the 1980s, actinomycetes were initially isolated from DQ and fermented grains from JXXB, and subsequently, isolation and identification of actinomycetes focused on the PM and DQ used for making NXXB [22,23]. A total of 86 species of actinomycetes have been isolated from five types of Baijiu (JXXB, NXXB, QXXB, ZMXXB, and FXXB), from different regions, and from the DQ, PM, and FGs. The major genera of actinomycetes isolated from Baijiu include *Thermoactinomyces*, *Streptomyces*, *Nocardiopsis*, *Massilia*, *Shimazuella*, *Kroppenstedtia*, *Laceyella*, *Micromonospora*, and *Arthrobacter*. Summary information on the previously isolated and identified actinomycetes from different stages of Baijiu production is shown in Table 1, and more detailed information on the growth media in Appendix A.

*Streptomyces albus* was detected in a cellar where JXXB is made, the PM used forNXXB, and the DQ used in FXXB and QXXB [28,39,50,54]. *Streptomyces cacaoi* was isolated from JXXB, NXXB, and QXXB [26,36,50]. *Thermoactinomyces vulgaris*, *Laceyella sacchari*, and *Thermoactinomyces intermedius* were isolated from the high-temperature DQ used for JXXB and ZMXXB [24,48]. *Streptomyces griseus*, *Streptomyces zaomyceticus*, *Thermoactinomyces thalpohillus*, and *Streptomyces flocculus* were isolated from JXXB DQ, using Gao’s No. 1 medium [26,28,32]. *Aggregatibacter actinomycetemcomitans*, *Thermostaphylospora chromogena*, *Streptomyces flocculus*, and *Streptomyces* sp. R11-21 were isolated from FG [31,32,33,34]. *Streptomyces bangladeshensis* and *Streptomyces rochei* were isolated from DQ, using GTY medium and casein medium, respectively [25,30].

Gao’s No. 1 is a commonly used medium for isolating actinomycetes from NXXB fermentations. Seven, two, and twelve species were isolated from DQ, FGs, and PM, respectively [36,43]. *Arthrobacter protophormiae* was isolated using inorganic salt medium from 20-year-old PM [38]. *Streptomyces sampsonii* and *Streptomyces rutgersensis* were separated from the in situ medium and the *Thermophilibacter gallinarum* R2A medium from the PM [41,42]. Three genera (*Streptomyces*, *Nocardiopsis*, and *Massilia*) and 26 species were isolated from the fermentations using modified Gao’s No.1 medium [35]. Ten species of *Streptomyces* were isolated using Gao’s No.1 medium and two using HV medium from the DQ and FG of QXXB; *Streptomyces* ZYP11, *Streptomyces* ZYP9, *Streptomyces* ZYP8, and *Brevibacterium renqingii* were also isolated from QXXB using GW1, R2A, GMKA, and LSA media, respectively [15,49]. *Shimazuella kribbensis* was only found in the FG from QXXB fermentation [52]. *Thermoactinomyces daqus* H-18 and *Streptomyces thermoviolaceus* were isolated using R2A medium from ZMXXB fermentation [48,56]. *Arthrobacter aresens* was only found in the DQ and FG of FXXB [53,55]. Using BiologEcoPlates, 13 actinomycetes were isolated from 50-year-old PM used for NXXB [44].

## 3. Methods for Identification of New Species of Microorganisms from Baijiu Production

Culture-independent methods, such as polymerase chain reaction denaturing gradient gel electrophoresis (PCR-DGGE) and sequencing technology, have been developed for identification of actinomycetes (Table 2). The microbiomes of four different types of Baijiu were analyzed by these methods. *Thermoactinomyces sanguinis* could only be detected in the DQ and FG of JXXB, NXXB, QXXB, and JXXB by culture-independent methods [57,58,59,60]. *Thermoactinomyces vulgaris*, *Olsenellauli*, *Olsenella profusa*, *Lancefieldella parvula*, *Corynebacterium tuberculostearicum*, *Corynebacterium minutissimum*, *Streptomyces coeruleorubidus*, *Streptomyces hainanensis*, and *Arthrobacter woluwensis* were identified in NXXB fermentations by PCR-DGGE [61]. *Arthrobacter stackebrandtii*, *Kocuria carniphila*, *Glutamicibacter creatinolyticus*, *Brevibacterium aurantiacum*, *Cellulosimicrobium funkei*, *Microbacterium oxydans*, *Corynebacterium glutamicum*, *Gordonia terrae*, *Dietzia maris*, *Acidipropionibacterium acidipropionici*, *Microbacterium hydrocarbonoxydans*, *Microbacterium schleiferi*, and *Gulosibacter molinativorax* were first identified in the PM of NXXB by 16S rRNA gene sequencing [62]. *Streptomyces albus*, *Kroppenstedtia eburnea*, *Saccharopolyspora rectivirgula*, *Brevibacterium celere*, and *Thermoactinomyces vulgaris* were identified as the dominant microorganisms in the DQ of QXXB, according to a similar analysis of 16S rRNA [60]. In ZMXXB, *Saccharopolyspora rectivirgula*, *Saccharopolyspora hordei*, *Saccharopolyspora rectivirgula*, and *Rothiakristinae* were identified using 16S rDNA analysis [63]. *Thermoactinomyces vulgaris*, *Thermoactinomyces intermedius*, and *Thermoactinomyces daqus* were detected using ARDRA and each type represents an OTU (operational taxonomic unit) [64]. In addition, *Laceyellatengchongensis*, *Laceyellasediminis*, *Laceyellasacchari*, and *Laceyella putidus* were identified by a specific PCR assay using a new specific primer (109F/801R) [47].

Acomparison of actinomycetes identified by culture and culture-independent methods is shown in Appendix A. *Thermoactinomyces vulgaris*, *Streptomyces albus*, *Thermoactinomyces intermedius, Thermostaphylospora chromogena*, *Kroppenstedtia eburnea*, and *Streptomyces cacaoi* were identified by both methods. Eighty species were detected and identified by culture-independent methods, of which 55 species were detected without isolation. There are four major differences between the two methods: (i) the analyzed samples were from many different locations and sampling times; (ii) the isolation medium of actinomycetes were limited by traditional culture methods; (iii) metagenomics analysis of most samples by the cultured-independent methods could only assign the taxonomic classification of actinomycetes to genera; and (iv) low-abundance actinomycetes cannot be detected by targeted metagenomics such as 16S rRNA amplicon sequencing. To understand the diversity of actinomycetes within the Baijiu microbiome of a limited time, multi-omics analysis and more separation methods should be applied [75,76]. Further, combined with innovative techniques (high-throughput culture), uncultured actinomycetes should be targeted for isolation according to their special functional characteristics [77]. With the help of high-throughput analysis results, selective nutrient media, selective physicochemical conditions, density-based separation, inhibitors, and specific growth factors could be found to isolate the uncultured actinomycetes from the DQ, FGs, and PM of the different Baijiu ecosystems [77].

## 4. Distribution of Actinomycete Species from the Major Types of Baijiu

The distribution of actinomycete species from the major types of Baijiu, and from the DQ, PM, and FGs, are shown in Figure 1, and a more detailed distribution of *Thermoactinomyces* and actinomycetes in DQ is shown in Figure 2. The variation in the profile of the actinomycetes was unraveled among the niches of the different types of Baijiu. *Thermoactinomyces*, a thermophilic actinomycete, was found in the DQ and FG of JXXB from three Baijiu factories, and accounted for 34.4–66.1% of the microbiome during the DQ production process [78,79,80]. Actinomycetes, including the *Actinopolysporaceae*, *Brevibacteriaceae*, and *Streptomycetaceae* families, make up 3.3–9.1% of the DQ microbiome [81]. Actinomycetes were one of the main phyla (7.7%) and *Thermoactinomyces* was the dominant genus in white and yellow DQ [82]. Three types of high-temperature DQ contained four genera of actinomycetes, namely, *Saccharopolyspora*, *Brevibacterium*, *Streptomyces*, and *Thermoactinomyces*, which comprised 4.7, 1.2, 1.5, and 26.4% of the total sequences, respectively [83]. *Thermoactinomyces* and *Saccharopolyspora* accounted for 16.7 and 22.1%, respectively, of the genera in Northern JXXB DQ, under high-temperature conditions [84]. The abundance of actinomycetes in manually and mechanically produced high-temperature DQ was 1.94 and 3.08%, respectively, and increased to 14.19 and 5.06% during fermentation [85]. JXXB goes through multiple rounds of fermentation, where one round includes three distinct phases: stacking fermentation, anaerobic fermentation, and distillation. *Thermoactinomyces* and *Streptomyces* have very low abundances at the early stage of fermentation, from the first to the seventh rounds, but they are dominant at the middle stage of the seventh round [66,86]. Fermented grains undergo three processes: cooling, stacking, and cellar fermentation. In the early stage of stacking fermentation, the FG microbiome contained 11.5% *Thermoactinomyces* [6].

The relative abundance of actinomycetes increased in NXXB DQ from Day 5 to Day 20 of fermentation, then decreased over the next few weeks during fermentation and maturation [87]. *Kroppenstedtia* and *Thermoactinomyces* were dominant in medium- and high-temperature DQ, accounting for 45.1% and 12.3%, respectively [88]. *Thermoactinomyces* was dominant during storage of DQ, whereas *Saccharopolypora* and *Micromonospora* increased in abundance during the subsequent fermentation [89,90]. *Saccharopolyspora* and *Thermoactinomyces* were the dominant genera after 18 days of DQ fermentation, reaching maximum abundances of 33.8% and 17.8%, respectively. The relative abundance at the phylum level for actinomycetes was 16.4% and that of *Thermoactinomyces* in mature DQ was 16.3% (Yibin) and 13.2% (Luzhou) [91]. The relative abundance of *Thermoactinomyces* reached 78% for special-grade DQ and 33% for first-grade DQ during summer production [92].

Dynamic changes in actinomycete relative abundance were studied in the DQ, PM, and FGs of NXXB [93]. During the fermentation and maturation of DQ, the relative abundance of actinomycetes initially increased, then decreased, and the relative abundance of the different species also changed, including *Thermoactinomyces*, *Saccharopolypora*, and *Micromonospora. Kroppenstedtia* and *Thermoactinomyces* were dominant in medium- and high-temperature DQ, accounting for 45.1% and 12.3%, respectively [88]. Two genera of actinomycetes, *Kroppenstedtia* and *Thermoactinomyces*, dominated the DQ at a medium and high temperature, accounting for 45.11% and 12.29%, respectively. *Saccharopolyspora* and *Thermoactinomyces* were the dominant genera after 18 days of DQ fermentation, reaching maximum abundances of 33.8% and 17.8%, respectively [94]. *Thermoactinomyces* in mature DQ from different locations accounted for 16.26% (Yibin) and 13.21% (Luzhou) [91]. Among the different grades of DQ, the proportion of *Thermoactinomyces* in special-grade DQ reached 78% and that of first-grade DQ reached 33% [92].

In the PM of NXXB, the relative content of actinomycetes initially decreased and then tended towards stability with increased cellar age [95,96]. The relative abundance of actinomycete genera, such as *Atopobium* and *Olsenella*, gradually increased by more than 1% during aging of PM and then stabilized in mature PM [97,98,99,100]. Moreover, the abundance of actinomycetes in mature PM (1.68 × 10^10^ copies per g) was 29 times that of aging PM (0.58 × 10^9^ copies per g) [101], and the matured PM (2.23 × 10^9^) was 24 times that of degraded PM (9.25 × 10^7^ cells/g) [102]. In the same cellar, the diversity of the microbiome in mature pit mud was superior to that in degraded pit mud; the relative abundance of the core actinomycete species, such as *Frankia casuarinae*, *Brachybacterium faecium*, and *Mycobacterium sinense* increased, because of the long-term anaerobic conditions in PM [103]. A study of actinomycetes in different layers of a Baijiu cellar detected actinomycetes at a relative abundance of >1% in the microbiome of the middle and upper layers [100,104], whereas the relative abundance of the actinomycetes was up to 4.81% in the bottom layer [105].

In FGs, the abundance of actinomycetes gradually decreases during fermentation. During the first week, *Actinobacter* and *Saccharopolyspora* were the dominant genera, at >2% [106]. Subsequently, the dominant actinomycetes were *Thermoactinomyces*, *Arthrobacter*, and *Corynebacterium*, accounting for more than 1% [107]. The relative abundance of actinomycetes was 1.78, 1.16, and 0.31% in the FGs after 3, 15, and 45 days, respectively [90].

In qingxiangxing Baijiu (QXXB), Daqu is classified into low temperature, medium temperature, and high temperature, according to the fermentation temperature used [108]. For a medium temperature (50–60 °C) and high temperature (60–70 °C) DQ, *Thermoactinomyces* was the dominant bacterium [109]. The low-temperature DQ of QXXB is further classified into three types, namely, QingCha (QC), HongXin (HX), and HouHuo (HH), each having a distinct production process [110]. Actinomycetes account for 73% in QC, 19% in HX, and 1% in HH [110]. *Thermoactinomyces* and *Streptomyces* reached a 19.5 and 14.1% relative abundance in HX, respectively, and the *Streptomyces* of QC reached 13.1% [110]. At the center and surface of the QC, HX, and HH DQ blocks, the actinomycete relative abundance was 7.6, 8.5, and 8.4% at the center and 6.9, 7.8, and 7.1% at the surface, respectively [111]. Metagenomics analysis showed that the relative abundance of actinomycetes in low-temperature Daqu (LTD) samples accounted for 24.25%, mainly *Streptomyces albus* (6.53%), *Streptomyces* sp. NHF165 (5.15%), and *Saccharopolyspora erythraea* (1.55%) [72]. There is also variation in actinomycete abundance during different storage periods of LTD. When LTD was stored for 6 months, actinomycetes accounted for 23.9, 17.9, and 1.1% in QC, HX, and HH, respectively, of which the dominant families included *Thermoactinomyces* (19.6% in HX, 2.1% in QC, and 1.8% in HH), and *Streptomyces* (14.1% in HX, 13.1% in QC, and 0.8% in HH) [110,112]. Comparing the interior and surface of LTD stored for two months, the dominant actinomycetes were *Thermoactinomyces vulgaris* in the interior and *Saccharopolyspora cebuensis*, *Brevibacterium* sp. D2, *Actinomyces* sp. 152R-3, and *Brachybacterium* sp. PB10 at the surface [113].

The process for making zhimaxiangxing Baijiu (ZMXXB) DQ consists of four stages: initial fermentation, ripening fermentation, drying, and storage [114]. Actinomycetes gradually increased up to a maximum of 8% during the initial fermentation, and then decreased during ripening and drying [114]. In mature DQ, the main actinomycete genera were represented by *Thermoactinomyces*, *Kroppenstedtia*, and *Saccharopolyspora*, accounting for 52, 17.3, and 17.3%, respectively [46,73]. In DQ matured for three months, *Actinopolyspora* and *Thermoactinomyces* accounted for 5.1 and 15.7%, respectively [115]. High-temperature ZMXXB DQ is classified into three types, depending on the storage location, namely, white for the upper layer (35–45 °C), yellow for the middle layer (45–55 °C), and black for the lower layer (55–65 °C) [116]. The dominant actinomycetes were *Thermoactinomyces* (58.6%) in white DQ, *Kroppenstedtia* (16.9%) in yellow DQ, and *Saccharopolyspora* (17.60%) in black DQ [116]. In ZMXXB FG during the early stage of fermentation, the content of actinomycetes was >5% [117].

In fengxiangxing Baijiu (FXXB) FGs, the actinomycete concentrations were, 440, 259, and 261 cells/g in the upper, middle, and lower layers [118], respectively, during two weeks of fermentation; actinomycetes exceeded 4% in the first three days, slowly decreased to 0.5% in the second week, and then dropped sharply and stabilized at 0.2% [119]. As discussed above, actinomycetes are a major constituent of the microbiomes of DQ, FGs, and PM; *Thermoactinomyces* and *Saccharopolypora* are the dominant genera in mature DQ in the major types of Baijiu [112,116,120,121]. Actinomycetes are vital for Baijiu production, contributing to the quality of the final product by producing important flavor compounds, including four esters and five alcohols [7,119]; actinomycetes in the FG are derived from the DQ and PM, making a major contribution to starch saccharification. The niche adaptation and distribution of actinomycetes in brewage environments changes with some environmental factors such as temperature, moisture, and nutrient availability. Baijiu product quality and safety are associated with the brewing microbial community, which also depends on the actinomycete species. An improved understanding of the dynamic changes in actinomycete relative abundance under different fermentation conditions will require further research.

**Figure 2 foods-11-03551-f002:**
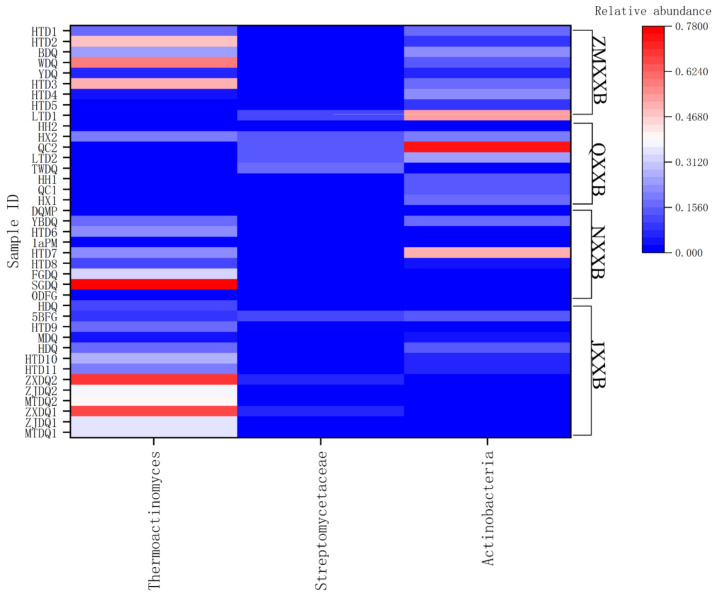
Relative abundance heatmap of *Thermoactinomyces* and actinomycetes in DQ samples obtained from high-temperature Daqu (HTD), white Daqu (WDQ), yellow Daqu (YDQ), black Daqu (BDQ), low-temperature Daqu (LTD), Taiwanese Daqu (TWDQ), the Daqu-making process (DQMP), Yibin Daqu (YBDQ), 1-year-old pit mud (1aPM), first-grade Daqu (FGDQ), special-grade DQ (SGDQ), 0-day fermented grain (0DFG), Hongtudi distillery Daqu (HDQ), 5th fermentation round FG (5BDQ), M-type Daqu (MDQ), H-type Daqu (HDQ), Zhongxin Daqu(ZXDQ), Zhenjiu Daqu (ZJDQ), and Maotai DQ (MTDQ) [6,63,71,72,73,79,80,82,83,84,85,88,91,92,95,107,110,112,114,116,122].

## 5. Interspecies Interactions of Actinomycetes and Other Microorganisms in Baijiu Fermentation

Baijiu flavor compounds are products of co-fermentation by multiple microorganisms. The interspecific interactions between actinomycetes and other microorganisms are closely related to the major flavor compounds in DQ, FGs, and PM [123,124]. Interspecies interactions between actinomycetes fall into four main categories (Figure 3). Actinomycetes produce various enzymes, such as cellulase, amylase, pectinase, and protease, which mediate these interactions, and the enzymolysis products can be assimilated by other microorganisms in the Baijiu fermentation microbiome [125]. For example, *Streptomyces avicenniae* hydrolyzes starch and produces melanin that scavenges free radicals from the cell surface of *C. butyricum*, promoting growth and caproic acid production [19,126] (Figure 3A). Some actinomycete metabolites are precursors for flavor component production by the key microorganisms *Bacillus* and caproic acid-producing bacteria (CPB) [18,37]. Acetic acid and lactic acid produced by *Streptomyces* and *Micromonospora* are precursors for yeast or *Bacillus* to produce ethyl acetate and ethyl lactate [37] (Figure 3B). Actinomycetes inoculated into CPB fermentation medium promote production of caproic acid and ethyl caproate [127]. Co-culture of actinomycetes with lactic acid-producing bacteria (LPB) or acetic acid-producing bacteria (APB) promote the growth of the latter and production of ethyl lactate and ethyl acetate, which improves the Baijiu quality [18]. Actinomycetes produce antibiotics that inhibit pathogenic and functional bacteria, for example, 15 strains of *Streptomyces* could inhibit human pathogenic bacteria during DQ production [15]. These strains of *Streptomyces* both produce heptaene macrolide antibiotics, which inhibit the growth of yeasts, and degrade alcohols (3-octanol and 3-methyl butanol) and esters (ethyl octanoate and ethyl decanoate) [128] (Figure 3C). Non-protein and non-peptide antibiotics, or quinomycinA produced by actinomycetes, inhibit the biological activity of *Bacillus subtilis* [129,130]. In DQ production, *Bacillus* strains inhibit the growth of *Streptomyces* and degrade the geosmin produced by *Streptomyces* (Figure 3D) [131]. Therefore, *Bacillus amyloliquefaciens* reduces the concentration of geosmin and the growth of *Streptomyces* strains, which relieves the inhibition of pyrazine compound production by *Streptomyces* and inhibits the formation of off-odors [132]. With further research, *Bacillus subtilis* may be able to downregulate the gene expression of the streptomycin *Streptomyces griseus*, which reduces the inhibiting effect of the latter [133].

The interaction mechanism between different actinomycetes and with other microorganisms is still poorly understood, but there is great potential for improving our understanding of this using ‘omics technology [75,134]. Moreover, genome-scale metabolic models (GSMMs), which can analyze and visualize the interaction mechanisms between actinomycetes and functional microorganisms in the Baijiu, should be constructed and applied [135].

## 6. Actinomycete ‘Omics Research

The important metabolic pathways and functions of actinomycetes isolated from Baijiu fermentations have been analyzed by sequencing and annotation of the genomes of several actinomycete species (Table 3). Annotation of the *Thermoactinomyces daqus* H-18 genome revealed 1184 enzymatic reactions, 264 transporters, 867 compounds, 2361 transcription units, and 6 coding sequences (CDSs) of heat-shock proteins, which confer tolerance of high temperatures [45]. During growth at 60 °C in high-temperature DQ, actinomycete gene expression related to fatty acid biosynthesis increased six-fold [136]. The genes *Clp*, *groEL*, and *pstB* in thermophilic actinomycetes increase their survivability at a high temperature, allowing them to become dominant when increasing the DQ temperature [64]. Genome annotation of *Streptomyces* sp. FBKL4.005 revealed the genes coding for metabolic pathways associated with characteristic Baijiu flavors, sugar degradation, and streptomycin and neomycin production [29].

Comparative genomics has revealed the wide diversity of gene clusters in actinomycetes [137]. Genes coding for the geosmin biosynthetic pathway were all found in the core genome of *Streptomyces* [138]. In addition, metaproteomics and metabolomics were also used to investigate the functional changes in actinomycetes [68,139,140]. Actinomycete abundance had a negative correlation with lactic acid and a positive correlation with pH, determined by transcriptomic sequencing [141]. Actinomycete transcriptomic analysis revealed that aged PM has a higher content of seven key enzymes than degraded PM [103].

The combination or comparison of multiple ‘omics studies has not yet been applied to actinomycetes from the Baijiu microbiome to analyze and amplify their genetic elements and metabolic pathway. The GSMMs of Baijiu actinomycetes have not yet been constructed and analyzed. ‘Omics analysis, combined with the GSMMs of actinomycetes, should be performed to attain a comprehensive understanding of actinomycetes in future research. Gene engineering could be used to regulate and produce natural products from Baijiu actinomycetes.

## 7. Enzymes and Metabolites Produced by Baijiu Actinomycetes

It was found that actinomycetes are characterized by metabolic diversity in the following. Actinomycetes have been found to produce cellulase, amylase, pectinase, protease, endoglucanase, and beta-glucosidase, which can contribute to the saccharification and fermentation of Baijiu [15,19,123,142]. During fermentation, the lipase and phosphatase produced by *Thermoactinomyces* can reduce the ethyl lactate production [116]. *Thermoactinomyces vulgaris* also produces protease, which can hydrolyse protein into amino acids [46]. Aside from enzymes, actinomycetes also produce many metabolites, including aroma compounds, antibiotics, and off-odor compounds (Table 4). *Thermophilibacter gallinarum* can convert glucose into lactic acid and acetic acid [41]. Actinomycetes also have the potential to produce butyric acid, thereby providing the precursor for ethyl butyrate and enriching the flavor of the Baijiu [143]. *Thermoactinomyces* can use a wide variety of carbon and nitrogen sources and synthesize important flavor components, such as ethyl caproate, furfuryl alcohol, phenethyl alcohol, pyrazine, butanoate, and acetic acid [143]. *Streptomyces* species synthesize some flavor compounds, such as 3-hydroxy-2-butanone, 2,3-butanediol, ethanol and ethyl acetate, butanol, acetone, 3-methyl-3-buten-1-ol, and dimethyl disulfide [39,42,144]. *Thermoactinomyces* and *Actinomycetales* produce 5 pyrazines (trimethylpyrazine, tetramethylpyrazine, 2,3,5-trimethyl-6-ethylpyrazine, 2,5-dimethyl-3-butylpyrazine and 2,5-dimethyl-3-(3-methylbutyl)pyrazine), 2 aromatics (4-ethenyl-1,2-dimethoxy-benzene and (1-pentylhexyl)-benzene), and 1 alcohol (1-octen-3-ol) [145]. *Streptomyces* sp. R11-21 growing on wheat can produce 2-methyl isobornyl alcohol and terpenoid compounds, and 3-hydroxy-2-butanone and terpenes when growing on sorghum [33]. *Streptomyces bangladeshensis* is capable of producing terpenes and pyrazine [30]. *Streptomyces* in NXXB fermentations can produce esters (ethyl butyrate, ethyl lactate, and ethyl caproate), acids (butyric acid and caproic acid), and aldehydes (acetaldehyde and furfural) [146]. More meaningfully, the main actinomycetes strains were capable of producing antibiotics [26,27,147]. *Streptomyces aureus* and *Streptomyces albus* can produce the brown pigments salinomycin and polylysine, respectively, which inhibit the growth of functional bacteria and form the main aroma compounds [148,149,150]. Four strains of *Streptomyces* secrete heptane macrolide antibiotics, which inhibit fungal growth [128]. Actinomycetes can produce antibiotics that inhibit harmful microorganisms [151]; for example, *Streptomyces* strains that produce geosmin, which are regarded as serious microbial contaminants of the PM [16].

## 8. Potential Applications of Actinomycetes in Baijiu Fermentation

Actinomycetes have useful regulatory functions in Baijiu fermentation, as discussed below. Actinomycetes produce highly active hydrolytic enzymes that enable full utilization of all the components of baijiu FGs [15,153], particularly cellulase, which degrades the abundant cellulose to produce short-chain fatty acids in FGs and distiller’s grains [154]. Undesirable isopropanol and lactic acid produced in Baijiu fermentation are degraded by *Arthrobacter protophormiae* [38,155], and actinomycetes produce antibiotics that inhibit the growth of human pathogenic bacteria [15]. Geosmin produced by *Streptomyces* ameliorates the effect of excess acidity during fermentation and since actinomycetes are relatively heat tolerant, they can maintain their metabolic activity in high-temperature DQ and PM [128,156]. The hyphae of *Thermoactinomyces* facilitate the evaporation of water from DQ and help to soften the grains, which aids their starter function [128]. Actinomycetes, as dominant strains in PM, are regarded as indicators of PM aging, and are studied to distinguish between the PM of different maturities [4,96]. Actinomycetes can facilitate denitrification of the PM using sulfur and sulfides and inhibit degradation of the PM [157]. New DQ and PM inoculated with selected actinomycetes reach maturity relatively quickly and old PM can be maintained in good condition, thereby maintaining Baijiu quality [102]. As stated above, actinomycetes are important for DQ biocontrol, PM maintenance, the formation of Baijiu flavor, and resource utilization of distiller’s grains. However, for the future practical application of actinomycetes, some progress needs to be made to promote caproic acid production, inhibit the growth of pathogenic bacteria, and degrade distiller’s grains. Hence, actinomyces has the potential to improve the safety and quality of Baijiu, which provide a direction for applied value research.

## 9. Conclusions

Actinomycetes hydrolyze starch, protein, and cellulose to supply precursors for other microorganisms to produce flavor components during fermentation, as well as producing important Baijiu flavor compounds, such as ethyl caproate and ethyl butyrate. Actinomycete relative abundance and species distribution are used as an indicator of PM quality and are important microorganisms for inhibition of PM degradation. However, the metabolic mechanism, isolation methods, and underlying interactions between actinomycetes are poorly understood, and in-depth research on the multi-omics analysis of actinomycetes has not yet been reported. Therefore, innovative separation methods are needed to efficiently isolate unculturable actinomycetes from complex microbiomes. The omics-based approaches enhanced our understading of the diversity and functional dynamics of actinomycetes. Future studies should also consider the potential application of actinomycetes to improve the food safety of Baijiu during fermentation, by regulating harmful microorganisms, which would improve Baijiu quality. Future research should include multi-‘omics studies and construction of actinomycete GSMMs, by combining bioinformatics tools, high-throughput culture methods, and genetic engineering. From the fundamental basis and new insights into actinomycetes that were provided here, through an in-depth theoretical study of Baijiu microbial populations, future studies can help improve the Baijiu product safety, sustainability, and brewage standards.

## Figures and Tables

**Figure 1 foods-11-03551-f001:**
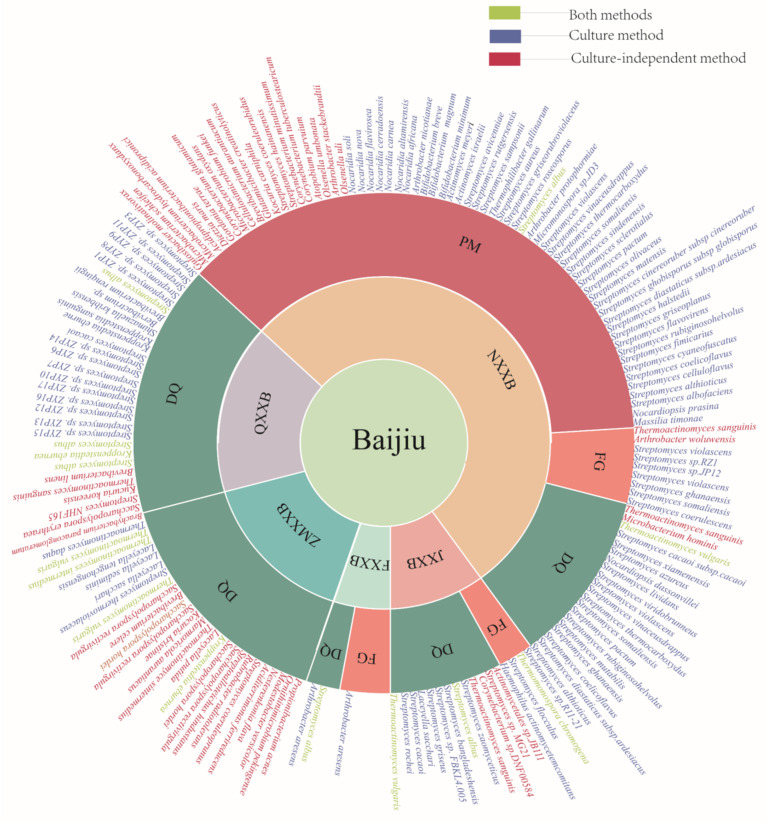
Actinomycete species distribution in the DQ (Daqu), FGs (fermented grains), and PM (pit mud) from different types of Baijiu—jiangxiangxing (JXXB), nongxiangxing (NXXB), qingxiangxing (QXXB), zhimaxiangxingbaijiu (ZMXXB), and andmixiangxing (MXXB).

**Figure 3 foods-11-03551-f003:**
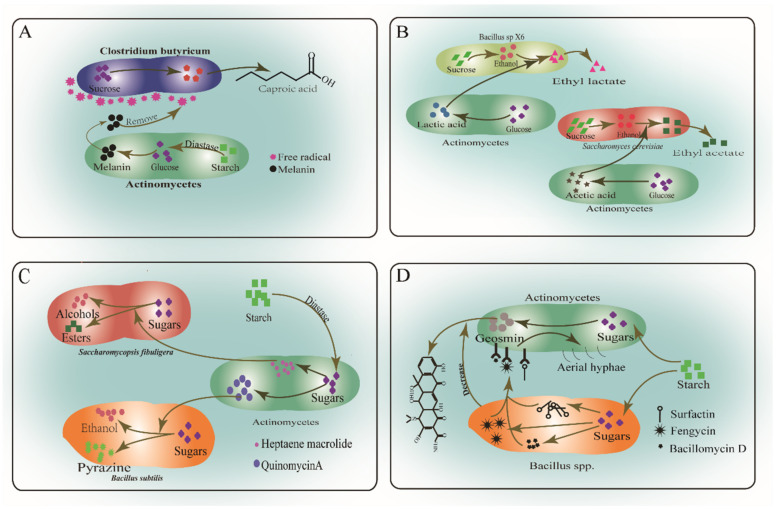
Interspecies interactions of actinomycetes. (**A**) Actinomycetes produce diastase to convertstarch into melanin that scavenges free radicals around *C. butyricum*. (**B**) Acetic acid and lactic acid produced by actinomycetes promote ethyl caproate and ethyl lactate production by yeasts and *Bacillus*. (**C**) Antibiotics produced by actinomycetes inhibit yeasts, promoting alcohol and pyrazine production by *Bacillus*. (**D**) *Bacillus* produces lipopeptides that inhibit the growth of *Streptomyces* and reduce geosmin production.

**Table 1 foods-11-03551-t001:** Actinomycete species isolated and identified from Baijiu.

Samples	Places	Types	Species	Media	References
High-temperature DQ ^1^	Maotai, Guizhou	JXXB	*Thermoactinomyces vulgaris*	GTY medium	[24]
High-temperature DQ	Sichuan	JXXB	*Streptomyces rochei*	Casein medium	[25]
High-temperature DQ	Guizhou	JXXB	*Streptomyces cacaoi*, *Streptomyces zaomyceticus*	Gause No. 1 medium, ISP2 medium	[26]
High-temperature DQ	Maotai, Guizhou	JXXB	*Laceyella sacchari*	Modified Gause No. 2 Medium, ISP2 medium	[27]
High-temperature DQ	Guizhou	JXXB	*Streptomyces griseus*, *Streptomyces albus*	Gause No. 1 medium	[28]
High-temperature DQ	Maotai, Guizhou	JXXB	*Streptomyces* sp. FBKL4.005	ISP2 medium	[29]
High-temperature DQ	Maotai, Guizhou	JXXB	*Streptomyces bangladeshensis*	GTY medium	[30]
FG ^2^	Gulin, Sichuan	JXXB	*Aggregatibacter actinomycetemcomitans*	Beef extract peptone medium	[31]
Alcoholic fermentative material	Huaihua, Guizhou	JXXB	*Streptomyces flocculus*	Gause No. 1 medium	[32]
Soilof baijiu production environment	Guizhou	JXXB	*Streptomyces* sp. R11-21	Gause No. 1 medium	[33]
FG	Huaihua, Guizhou	JXXB	*Thermostaphylospora chromogena*	Gause No. 1 medium	[34]
DQ	Yibin, Sichuan	NXXB	*Streptomyces althioticus*, *Streptomycescoelicoflavus*, *Streptomyces diastaticus* subsp. *ardesiacus*, *Streptomyces ghanaensis*, *Streptomyces mutabilis*, *Streptomyces pactum*, *Streptomyces rubiginosohelvolus*, *Streptomyces somaliensis*, *Streptomyces thermocarboxydus*, *Streptomyces vinaceusdrappus*, *Streptomyces violascens*, *Streptomyces viridobrunneus*	Modified Gause No. 1 Medium	[35]
FG	Yibin, Sichuan	NXXB	*Streptomyces coerulescens*, *Streptomyces somaliensis*, *Streptomyces ghanaensis*, *Streptomyces violascens*,		
PM ^3^	Yibin, Sichuan	NXXB	*Massilia timonae*, *Nocardiopsis prasina*, *Streptomyces albofaciens*, *Streptomyces althioticus*, *Streptomyces celluloflavus*, *Streptomyces cinereoruber* subsp. *cinereoruber*, *Streptomyces coelicoflavus*, *Streptomyces cyaneofuscatus*, *Streptomyces diastaticus* subsp. *ardesiacus*, *Streptomyces fimicarius*, *Streptomyces flavovirens*, *Streptomyces ghobisporus* subsp. * globisporus*, *Streptomyces griseoplanus*, *Streptomyces halstedii*, *Streptomyces matensis*, *Streptomyces olivaceus*, *Streptomyces pactum*, *Streptomyces rubiginosohelvolus*, *Streptomyces sclerotialus*, *Streptomyces sindenensis*, *Streptomyces somaliensis*, *Streptomyces thermocarboxydus*, *Streptomyces vinaceusdrappus*, *Streptomyces violascens*		
Mature medial-temperature DQ	Henan	NXXB	*Streptomyces lividans*, *Nocardiopsis dassonvillei*, *Streptomyces azureus*, *Streptomyces xiamenensis*, *Streptomyces cacaoi* subsp. *Cacaoi*	Gause No. 1 medium	[36]
Sealing mud	Yibin, Sichuan	NXXB	*Streptomyces* sp. JP12	Gause No. 1 medium	[37]
FG	Yibin, Sichuan	NXXB	*Streptomyces* sp. RZ1		
PM	Yibin, Sichuan	NXXB	*Micromonospora* sp. JD3		
The air of fermentation pit	Yibin, Sichuan	NXXB	*Streptomyces vinaceusdrappus*,		
PM (20 years)	Yibin, Sichuan	NXXB	*Arthrobacter protophormiae*	enriched medium and Inorganic salt medium	[38]
PM	Yibin, Sichuan	NXXB	*Streptomyces albus*	Gause No. 1 medium	[39]
PM	Luzhou, Sichuan	NXXB	*Streptomyces roseosporus*, *Streptomyces griseorubroviolaceus*, *Streptomyces aureus*	Gause No. 1 medium and complete Inorganic Basal Medium	[40]
PM	Sichuan	NXXB	*Thermophilibacter gallinarum*	R2A medium	[41]
PM (50 years)	Southern of Sichuan	NXXB	*Streptomyces sampsonii*, *Streptomyces rutgersensis*	Situ-medium	[42]
PM	Hubei	NXXB	*Streptomyces avicenniae*	Gause No. 1 medium	[43]
PM (50 years)	Anhui	NXXB	*Actinomyces israelii*, *Actinomyces meyeri*, *Bifidobacterium minimum*, *Bifidobacterium magnum*, *Bifidobacterium breve*, *Arthrobacter nicotianae*, *Nocardia africana*, *Nocardia altamirensis*, *Nocardia carnea*, *Nocardia cerradoensis*, *Nocardia flavirosea*, *Nocardia nova*, *Nocardia xishanensis*	Isolation medium	[44]
DQ	Zibo, Shandong	ZMXXB	*Thermoactinomyces daqus*	R2A medium	[45]
High-temperature DQ	Shandong	ZMXXB	*Thermoactinomyces vulgaris*	R2A medium	[46]
High-temperature DQ	Shandong	ZMXXB	*Thermoactinomyces intermedius*, *Laceyella tengchongensis*, *Laceyella sediminis*, *Laceyella sacchari*, *Laceyella putida*	ISP2 medium	[47]
High-temperature DQ	Shandong	ZMXXB	*Thermoactinomyces vulgaris*, *Streptomyces thermoviolaceus*	R2A medium	[48]
DQ	Xinghuacun, Shanxi	QXXB	*Streptomyces* sp. ZYP3, *Streptomyces* sp. ZYP6, *Streptomyces* sp. ZYP7, *Streptomyces* sp. ZYP10, *Streptomyces* sp. ZYP12 *Streptomyces* sp. ZYP13, *Streptomyces* sp. ZYP15, *Streptomyces* sp. ZYP16, *Streptomyces* sp. ZYP17, *Streptomyces* sp. ZYP18	GS medium	[15]
DQ	Xinghuacun, Shanxi	QXXB	*Streptomyces* sp. ZYP11	GW1 medium	
DQ	Xinghuacun, Shanxi	QXXB	*Streptomyces* sp. ZYP9	R2A medium	
DQ	Xinghuacun, Shanxi	QXXB	*Streptomyces* sp. ZYP8	GMKA medium	
DQ	Xinghuacun, Shanxi	QXXB	*Streptomyces* sp.ZYP1, *Streptomyces* sp. ZYP14	HV medium	
DQ	Shanxi	QXXB	*Brevibacterium renqingii*	LSA medium	[49]
DQ	Beijing	QXXB	*Streptomyces albus*, *Streptomyces cacaoi*	ISP2 medium	[50]
Out part of DQ	Shanxi Xinghuacun	QXXB	*Bervibactrium* sp. *Micrococcus lutens*	MRSA medium	[51]
DQ, FG	Beijing	QXXB	*Shimazuella kribbensis*, *Kroppenstedtia sanguinis*, *Kroppenstedtia eburnea*	Modified Gause No. 2 Medium	[52]
DQ	Shaanxi	FXXB	*Arthrobacter aresens*	Gause No. 1 medium	[53]
DQ	Shaanxi	FXXB	*Streptomyces albus*	PDA medium	[54]
FG	Shaanxi	FXXB	*Arthrobacter aresens*	Gause No. 1 medium	[55]

^1^ Daqu; ^2^ fermented grain; ^3^ pit mud.

**Table 2 foods-11-03551-t002:** Actinomycete species identified by culture-independent methods.

Samples	Types	Places	Methods	Species	References
High-temperature DQ ^1^	JXXB	Sichuan	PCR-DGGE	*Thermoactinomyces sanguinis*	[57]
DQ	JXXB	Jiangsu	PCR-DGGE	*Thermoactinomyces sanguinis*	[65]
FG ^2^	JXXB	Guizhou	PCR-DGGE	*Thermoactinomyces sanguinis*	[58]
FG	JXXB	Renhuai, Guizhou	PCR-DGGE	Uncultured *Actinomycete* colone 4-306, *Corynebacterium* sp. *DNF00584, Streptomyces* sp. MG21, *Actinomycetales* sp. JB111, *Thermoactinomyces sanguinis*	[66]
DQ	NXXB	South	PCR-DGGE	*Thermoactinomyces vulgaris*	[67]
Baobaoqu	NXXB	Sichuan	LC-MS/MS	*Microbacterium hominis*, *Thermoactinomyces vulgaris*	[68]
High-temperature DQ	NXXB	Anhui	16S rDNA	*Thermoactinomyces sanguinis*	[59]
PM ^3^	NXXB	Sichuan	PCR-DGGE	*Olsenella uli*, *Olsenella profusa*, *Lancefieldellaparvuium*, *Corynebacterium tuberculostearicum*, *Corynebacterium minutissimum*, *Streptomyces coeruleorubidus*, *Streptomyces hainanensis*	[61]
PM	NXXB	Sichuan	16S rRNA	*Arthrobacter stackebrandtii*, *Kocuria carniphila*, *Glutamicibacter creatinolyticus*, *Brevibacterium aurantiacum*, *Cellulosimicrobium funkei*, *Microbacterium oxydans*, *Corynebacterium glutamicum*, *Gordonia terrae*, *Dietzia maris*, *Acidipropionibacterium acidipropionici*, *Microbacterium hydrocarbonoxydans*, *Microbacterium schleiferi*, *Gulosibacter molinativorax*	[62]
FG	NXXB	Sichuan	PCR-DGGE	*Arthrobacter woluwensis*	[69]
Alcoholic fermentative materials	NXXB	Sichuan	PCR-DGGE	*Thermoactinomyces sanguinis*	[70]
Low-temperature DQ	QXXB	Beijing, Shanxi, Taiwan, Heilongjiang, Hebei	16S rDNA	*Streptomyces albus*, *Kroppenstedtia eburnea*	[71]
Mature DQ	QXXB	Shanxi	16SrRNA	*Thermoactinomyces sanguinis*, *Streptomyces albus*, *Brevibacterium linens*, *Brachybacterium paraconglomeratum*, *Rothia koreensis*	[60]
DQ	QXXB	Xiangyang, Hubei	WGS	*Streptomyces albus*, *Streptomyces* NHF165, *Saccharopolyspora erythraea*	[72]
High-temperature DQ	ZMXXB	Shandong	16S rDNA	*Thermoactinomyces vulgaris*	[73]
DQ (8 days of fermentation)	ZMXXB	Shandong	16S rDNA	*Saccharopolyspora rectivirgula*, *Brevibacterium celere*	[63]
DQ (24 days of fermentation)	ZMXXB	Shandong	16S rDNA	*Saccharopolyspora hordei*, *Saccharopolyspora rectivirgula*, *Rothiakristinae*, *Marmoricola aurantiacus*	
DQ (49 days of fermentation)	ZMXXB	Shandong	16S rDNA	*Thermoactinomycesintermedius*, *Thermoactinomyces vulgaris*, *Laceyella putida*, *Kroppenstedtia eburnea*, *Saccharopolyspora hordei*, *Saccharopolyspora rectivirgula*, *Streptoalloteichus hindustanus*, *Rubrobacter radiotolerans*, *Streptomyces coeruleoprunus*, *Aciditerrimonasferrireducens*, *Nesterenkonia flava*, *Modestobacter versicolor*, *Ornithinimicrobium pekingense*, *Cutibacterium acnes*	
DQ		Northern and southwestern	16S rRNA and 26S rRNA	*Saccharypolyspora rosea*, *Saccharopolyspora rectivirgula*, *Saccharopolyspora hordei*, *Saccharopolyspora spinosa*, *Streptomyces albus*, *Streptomyces cacaoi*, *Thermoactinomyces sanguinis*, *Thermoactinomyces vulgaris*, *Thermobisporabispora*, *Thermostaphylospora chromogena*, *Actinopolyspora erythraea*	[74]

^1^ Daqu; ^2^ fermented grain; ^3^ pit mud.

**Table 3 foods-11-03551-t003:** Genomic features of the actinomycetes isolated from Baijiu.

Species	Strain	NCBI Access No.	Size (Mb)	GC%	Proteins	rRNA Operons	tRNA Genes	References
* Thermoactinomyces daqus *	H-18	NZ_JPST01000000	3.44	48.8	3440	5	58	[45]
*Streptomyces mutabilis*	Z9A-32	HQ238326	7.83	71.6	6711	19	82	[35]
*Streptomyces vinaceusdrappus*	W8A-43	HQ238406	8.46	72.5	7332	5	67	[35]
* Streptomyces violascens *	S11A-6	HQ238298	8.92	70.5	7,754	24	71	[35]
* Streptomyces griseus *	A2	JX007982	8.55	72.2	6968	18	67	[28]
* Streptomyces albus *	NRRL B-2365	DQ026669.1	7.59	72.7	6166	4	59	[28]
* Thermoactinomyces vulgaris *	ATCC15734	AF089892	2.62	48	2590	21	72	[24]
*Thermophilibacter gallinarum*	LZLJ-2T	NZ_JADCJZ000000000.1	1.85	65.2	1599	6	49	[41]
* Streptomyces cacaoi *	NBRC12748	NR041061	8.57	73.4	6679	8	61	[50]
* Arthrobacter stackebrandtii *	NG3	MT269547.1	4.43	65.6	3683	15	54	[62]

**Table 4 foods-11-03551-t004:** Secondary metabolites produced by the actinomycetes isolated from Baijiu.

Microorganisms	Substrate	Product	References
*Streptomyces* sp. R11-21	Flour	2-Methyl isobornyl alcohol and terpenes	[33]
	Wheat or sorghum	3-Hydroxy-2-butanone and terpenes	
*Streptomyces bangladeshensis*	Glucose	Terpenes and pyrazine	[30]
*Streptomyces albus* *Streptomyces sampsonii* *Streptomyces rutgersensis*	Wheat bran	3-Hydroxy-2-butanone, 2,3-butanediol, ethanol and ethyl acetate	[39]
*Streptomyces mutabilis* *Streptomyces vinaceusdrappus* *Streptomyces coelicoflavus* *Streptomyces violascens*	Glucose	Butyric acid, hexanoic acid, ethyl butyrate, ethyl hexanoate, ethyl lactate and furfural	[152]
*Thermoactinomycetes* FBKL4.010	Wheat	Furfuryl alcohol, phenethyl alcohol, 2,6-dimethylpyrazine, and 2,3,5,6-tetramethylpyrazine	
*Streptomyces* sp. A22	Wheat bran	Ethyl hexanoate and phenethyl alcohol	[132]
*Streptomyces aureus*	Tyrosine	Brown-pigment	[149]
*Streptomyces albus* S5	Glucose	Salinomycin	[148]
*Streptomyces cacaoi* *Streptomyces zaomyceticus*	Sorghum	Lipids	[26]
*Laceyellasacchari*	Sorghum	Tetramethylpyrazine	[27]
*Streptomyces sampsonii* *Streptomyces rutgersensis*	Oat	3-Hydroxy-2-butanone and 2,3-butanediol	[42]
	Glucose	Ethyl caproate, and geosmin	
*Streptomyces fradiae*, *Streptomyces radiopugnans*, *Streptomyces sampsonii*, *Streptomyces albus*	Starch	Geosmin	[16]
*Streptomyces* spp.	Starch	Ethyl lactate and caproic acid	[146]
*Streptomyces avicenniae*	Starch	Melanin	[126]
*Thermoactinomyces* sp.	Starch	Butanoate, Aceticacid, hexyl ester	[143]
*Streptomyces sampsonii*	Starch	Heptaene macrolide antibiotics	[128]
*Streptomyces albus* *Streptomyces griseus*	Cellulose	Bultanol, acetone, 3-methyl-3-buten-1-ol, dimethyl disulfide	[144]
*Thermoactinomyces* sp.	Starch	Pyrazines, aromatics and and alcohol	[145]
*Actinomycetales* sp.			
*Thermophilibacter gallinarum*	Glucose	Lactic acid and acetic acid	[41]
*Streptomyces albus*	Glucose	Poly-lysine	[150]
*Actinomycetes*	DQ	Enzymes, pyrazine and aromatic substances	[123]
	FG	Salinomycin and terpenes	
	PM	Acids, esters and terpenes	

## Data Availability

Not applicable.

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
