# Peer review of "Systematic Review of Actinomycetes in the Baijiu Fermentation Microbiome"

_foods, 2022, doi:10.3390/foods11223551_

Round 1

Reviewer 1 Report

Manuscript titled “Systematic review of actinomycetes in the baijiu fermentation microbiome” is referring to the progress on actinomycete research related to baijiu fermentation, including their isolation and identification, distribution, interspecies interactions, systems biology and main metabolites produced.

In most of the case the review looks like a list. It is not so easy reading. In my opinion authors should report the information giving more details about what they found in the previous literature.

The introduction section should be implemented. Authors may refer, for example, to the differences in geographical distribution, climate, manufacturing technique, and raw materials, and how they can influence the final product, including the flavor. This can be interesting, considering that it can play an important role in the natural selection of fermentation microflora with high acid tolerance.

Paragraph 2. Also, in this case authors should implement the section. They refer to “Isolation and identification of Actinomycetes from different stages of baijiu production” but it is reported just a list of microorganisms and the media used for the cultivation. The identification is reported in the next paragraph.

Lines 132-133: “To understand the diversity of actinomycetes within the baijiu microbiome of a limited time, more separation methods should be applied” Authors may suggest any? Any references?

Lines 136-138: “Using selective nutrient media, density-based separation and inhibitors, uncultured actinomycetes could be isolated from the DQ, FG, and PM of different baijiu ecosystems”. Please explain and add references.

Lines 214-215: “In qingxiangxingbaijiu (QXXB), Daqu is classified into low temperature, medium temperature and high temperature, according to the fermentation temperature used.” Please add references.

Lines 217-219: “The low-temperature DQ of QXXB is further classified into three types, namely QingCha (QC), HongXin (HH), and HouHuo (HH) via distinct production processes.” Please add references.

Line 218: “HongXin (HH)” please correct in HX.

Lines 235-236: “The process for making zhimaxiangxingbaijiu (ZMXXB) DQ consists of four stages: initial fermentation, ripening fermentation, drying and storage”. Please add references

Lines 242-244: “High-temperature ZMXXB DQ is classified into three types, depending on the storage location, i.e., white for the upper layer (35-45℃), yellow for the middle layer (45-55℃) and black for the lower layer (55-65℃).” Please add references

Lines 256-261: “High-temperature ZMXXB DQ is classified into three types, depending on the storage location, i.e., white for the upper layer (35-45℃), yellow for the middle layer (45-55℃) and black for the lower layer (55-65℃).” Please add references

Lines 276-279: “Baijiu flavor compounds are products of co-fermentation by multiple microorganisms. The interspecific interactions between actinomycetes and other microorganisms are closely related to the major flavor compounds in DQ, FG and PM. Interspecies interactions between actinomycetes fall into four main categories.” Please add references

Lines 285-287: “Some actinomycete metabolites are precursors for flavor component production by the key microorganisms Bacillus and caproic acid producing bacterias (CPBs).” Please add references

Lines 304-307: “With further research, Bacillus subtilis may be able to down-regulate gene expression of streptomycin, which reduces production of the inhibitors by Streptomyces griseus”. Please reformulate

Lines 398-400: “New DQ and PM inoculated with selected actinomycetes reaches maturity relatively quickly and old PM can be maintained in good condition, thereby maintaining baijiu quality.” Please add references

Paragraph 8: “8. Potential applications of actinomycetes in baijiu fermentation” This section needs to be implemented by the authors

Conclusion: please remove references. Please revise the conclusion section

Author Response

The link with the field of the Special Issue (Advance and Future Challenges to Microbial Food Safety) should be clearer. Consequently, could you revise, specially, the title and the abstract, to give the manuscript a proper/clearer orientation to the microbial food Safety of the baijiu, please?

Answer: Thank you for your kind advice. We have a request to replace the research topic (Advance and Future Challenges to Microbial Food Safety) with the research topic (New Insights in Microbial Diversity of Fermented Foods). We ensure that the title and abstract of the manuscript match the research topic.

For your guidance, itemized answers to each reviewer’s comments are appended below and it is marked in blue in the manuscript (answers are in blue font).

Answers to the Reviewer(s)' Comments:

  1. In most of the case the review looks like a list. It is not so easy reading. In my opinion authors should report the information giving more details about what they found in the previous literature.

Answer: Thank you for your comments. The purpose of this manuscript is to review current progress on the studies of actinomycete in baijiu ecosystem. To make the manuscript more readable, three pictures were added according to the information derived from the related references. The four tables in the manuscript could let the readers more easily understand composition, products, and function of actinomycetes in baijiu ecosystem. The comparison of the actinomycetes species identified from culture methods and uncultured methods pointed out the insufficient of current culture work of actinomycetes. The four types of interspecies interactions of Actinomycetes and other microorganisms were summarized for the first time. In addition, we systematically reviewed the products and application of actinomycetes from baijiu ecosystem, which could be helpful for the future develop of this bacteria.

  1. The introduction section should be implemented. Authors may refer, for example, to the differences in geographical distribution, climate, manufacturing technique, and raw materials, and how they can influence the final product, including the flavor. This can be interesting, considering that it can play an important role in the natural selection of fermentation microflora with high acid tolerance.

Answer: Thank you for your kind advice. We pointed out in the introduction section that “In essence, the diversity and stability of microbial communities associated with high acid tolerance are directly influenced by factors of natural environmental conditions (geographical limitations, temperature, humidity, pH and climate), raw materials (sorghum or a mixture of wheat, barley, corn, rice and sorghum) and complex production processes” influenced the microbial community structure. And the influence of microbial communities was analyzed because the high complexity of the microbiome naturally selected has a potential to produce distinct flavors containing different trace components and alcohol content (35%-60%) caused by their underlying metabolism and interactions. We cited 2 relevant references in the revised manuscript (lines 36-43).

Corresponding references in the answers:

Tan, Y.; Du, H.; Zhang, H.; Fang, C.; Jin, G.; Chen, S.; Wu, Q.; Zhang, Y.; Zhang, M.; Xu, Y. Geographically associated fungus-bacterium interactions contribute to the formation of geography-dependent flavor during high-complexity spontaneous fermentation. Microbiol. Spectrum. 2022,22, e01844.

Wang, L. Research trends in Jiang-flavor baijiu fermentation: From fermentation microecology to environmental ecology. J. food sci. 2022,87, 1362-1374.

  1. Paragraph 2. Also, in this case authors should implement the section. They refer to “Isolation and identification of Actinomycetes from different stages of baijiu production” but it is reported just a list of microorganisms and the media used for the cultivation. The identification is reported in the next paragraph.

Answer: Thank you for your suggestions. The table 1 contained the current information of isolated actinomycetes, not only including the media and samples, but also the species identification information. We have renamed the table 1 and added " from different stages of baijiu production " information in lines 71-74. The identification of actinomycetes was based on the culture-dependent methods and the culture-independent methods, which includes validation after isolation and cultivation. (1) In paragraph 2, we mainly described the identification of common actinomycete species isolated from different types of Baijiu ecosystem by culture-dependent methods. And certain actinomycete species were isolated from different types of samples in pure culture methods using different media. (2) In the next paragraph, we focus on the identification of actinomycete species by culture-independent methods. Then combining both methods to analyze and compare differences in actinomycete species identification.

4.Lines 132-133: “To understand the diversity of actinomycetes within the baijiu microbiome of a limited time, more separation methods should be applied” Authors may suggest any? Any references?

Answer: Thank you for your comments. We have added some information and inserted 2 references in lines 142-143.

Corresponding references in the answers:

Pal, S.; Jana, A.; Mondal, K. C.; Halder, S. K. "Omics Approach to Understanding Microbial Diversity". in Biotechnological Advances for Microbiology, Molecular Biology, and Nanotechnology, eds Jyoti, R. R.; Rout, G. K.; Abinash, D(Apple Academic Press). 2022, 23-36.

Kumar, R. R.; Jadeja, V. J.; Shree, M.; Virani, N. A. Isolation of Actinomycetes: A Complete Approach. Int. J. Curr. Microbiol. Appl. Sci. 2016,5, 606-618.

5.Lines 136-138: “Using selective nutrient media, density-based separation and inhibitors, uncultured actinomycetes could be isolated from the DQ, FG, and PM of different baijiu ecosystems”. Please explain and add references.

Answer: Thank you. We have made corresponding information in the revised manuscript and inserted a reference in lines 145-149. Through high throughput analysis of results, we might found the selective nutrient media (e.g specific substrates), physicochemical culture conditions (e.g temperature, pH, salinity) and the addition of inhibitors (e.g antibiotics and toxic compounds) and specific growth factors (e.g amino acids and vitamins) to progressively isolate and culture the uncultured actinomycetes.

Corresponding references in the answers:

Lewis, W. H.; Tahon, G.;Geesink, P.; Sousa, D. Z.; Ettema, T. E. G. Innovations to culturing the uncultured microbial majority. Nat. Rev. Microbiol. 2020,19, 225-240.

6.Lines 214-215: “In qingxiangxingbaijiu (QXXB), Daqu is classified into low temperature, medium temperature and high temperature, according to the fermentation temperature used.” Please add references. Lines 217-219: “The low-temperature DQ of QXXB is further classified into three types, namely QingCha (QC), HongXin (HH), and HouHuo (HH) via distinct production processes.” Please add references.

Answer: According to your kind suggestion, we added one reference in lines 225-226 and one reference in lines 229-230.

Corresponding references in the answers:

Feng, J. T.; Lu, Z. M.; Shi, W.; Xiao, C.; Zhang, X. J.; Chai, L. J.; Wang, S. T.; Shen, C. H.; Shi, J. S.; Xu, Z. H. Effects of different culture temperatures on microbial community structure, enzymeactivity, and volatile compounds in Daqu. Chin.J. Appl. Environ. Biol. 2021,27, 760-767.

Li, Z. J.; Fan, Y. H.; Huang, X. N.; Han, B. Z. Microbial Diversity and Metabolites Dynamic of Light-Flavor Baijiu with Stacking Process. Ferment. 2022,8, 67.

7.Line 218: “HongXin (HH)” please correct in HX.

Answer: Thank you for your kind advice. We have replaced "HongXin (HH)" with " HongXin (HX) " in the revised manuscript.

8.Lines 235-236: “The process for making zhimaxiangxingbaijiu (ZMXXB) DQ consists of four stages: initial fermentation, ripening fermentation, drying and storage”. Please add references. Lines 242-244: “High-temperature ZMXXB DQ is classified into three types, depending on the storage location, i.e., white for the upper layer (35-45℃), yellow for the middle layer (45-55℃) and black for the lower layer (55-65℃).” Please add references. Lines 256-261: “High-temperature ZMXXB DQ is classified into three types, depending on the storage location, i.e., white for the upper layer (35-45℃), yellow for the middle layer (45-55℃) and black for the lower layer (55-65℃).” Please add references.

Answer: Thank you for your kind suggestion. We add corresponding references related to it in "Lines: 247-248" and " Lines: 254-255".

Corresponding references in the answers:

Xie, M. W.; Lv, F. X.; Ma, G. D.; Farooq, A.; Li, H. H.; Du, Y.; Liu, Y. High throughput sequencing of the bacterial composition and dynamic succession in Daqu for Chinese sesame flavour liquor. J. Inst. Brew. 2019,126, 98-104.

Wu, X. Y.; Jing, R. X.; Chen, W. H.; Geng, X. J.; Li, M.; Yang, F. Z.; Yan, Y. Z.; Liu, Y. High-throughput sequencing of the microbial diversity of roasted-sesame-like flavored Daqu with different characteristics. 3 Biotech. 2020,10, 502.

9.Lines 276-279: “Baijiu flavor compounds are products of co-fermentation by multiple microorganisms. The interspecific interactions between actinomycetes and other microorganisms are closely related to the major flavor compounds in DQ, FG and PM. Interspecies interactions between actinomycetes fall into four main categories.” Please add references. Lines 290-292: “Some actinomycete metabolites are precursors for flavor component production by the key microorganisms Bacillus and caproic acid producing bacterias (CPBs).” Please add references.

Answer: Thank you. We added two references in Lines 288-289 and two references in Lines 295-296.

Corresponding references in the answers:

Guo, X. W.; Fan, E. D.; Ma, B. T.; Li, Z. X.; Zhang, Y. X.; Zhang, Z. M.; Chen, Y. F.; Xiao, D. G. Research progress in functional bacteria in solid-state fermented Baijiu in China. Food Ferment. Ind. 2020,46, 280-286.

Zou, W.; Zhao, C.; Luo, H.-b. Diversity and Function of Microbial Community in Chinese Strong-Flavor Baijiu Ecosystem: A Review. Front. Microbiol. 2018,9, 671.

Ren, Y. M.; Dai, S.; Fan, L.; Wei, M.; Shang, L. H.; Xie, W.; Zhuang, M. Y.; Hou, M. Z. Study on the isolation of actinomycetes and its application in the production of Lu-type liquor. Liquor Mak. Sci. Thchnol. 1997,3, 13-15.

Shi, S.; Zhang, X.; Yang, K. Z.; Liao, Q. J.; Qiao, Z. W.; Zheng, J.; Liu, D. T. Preliminary Study on the Regulation Effect of Actinomycetes on Brewing Microorganisms. Liquor Mak. Sci. Thchnol. 2021,2, 17-20.

10.Lines 304-307: “With further research, Bacillus subtilis may be able to down-regulate gene expression of streptomycin, which reduces production of the inhibitors by Streptomyces griseus”. Please reformulate.

Answer: Thank you for your constructive advices. We have rephrased the meaning of this sentence to make it more readable in lines 316-317.

11.Lines 398-400: “New DQ and PM inoculated with selected actinomycetes reaches maturity relatively quickly and old PM can be maintained in good condition, thereby maintaining baijiu quality.” Please add references.

Answer: Thank you for your comments. We have added the reference in the revised manuscript in lines 414-415.

Corresponding references in the answers:

Liang, H. P.; Luo, Q. C.; Zhang, A. Y.; Wu, Z. Y.; Zhang, W. X. Comparison of bacterial community in matured and degenerated pit mud from Chinese Luzho• flavour liquor distillery in different regions. J. Inst. Brew. 2016,122, 48-54.

12.Paragraph 8: “8. Potential applications of actinomycetes in baijiu fermentation” This section needs to be implemented by the authors. Conclusion: please remove references. Please revise the conclusion section

Answer: Thank you. We have removed references according to your suggestion. Undesirable components in Baijiu can be degraded by actinomycetes. Actinomycetes are able to inhibit the growth of pathogenic bacteria during the fermentation of Baijiu, which can improve the safety and stability of Baijiu product. In study, the quality of the pit mud was found to be associated with actinomycetes. In other respects, actinomycetes also have potential for fertilizer and wine distiller’s grains applications. Therefore, promoting caproic acid production, inhibiting the growth of pathogenic bacteria, degrading distiller’s grains and pit mud maintenance warrant further study. And we made some adjustments to expand the view in lines 415-421 and 432-433. In addition, we also modified the conclusion according to your advice.

Reviewer 2 Report

The authors gave a comprehensive documentation of the actinomycetes found during baijiu fermentation. 

The work is overall fine.

Below some suggestions

line 28: please elaborate what is meant with aroma types

tables 1-2: inconsistent font, spacing issues, Brevibacterium typo, Shaanxi or Shanxi?, make sure names are widely used, e.g I could not find the names Nocardia soli and africana in literature. A lot of names are incorrectly spelled or there is no space between the genus and species name.

line 110: It is Brevibacterium celere

line 143: types of

table 2 at reference 64 it is colony and not colone

line 199-200: font issue with 10^7 and 1-^9

line 201: better is not the best word to describe the diversity

line 249: cells

line 292: the growth of the latter

line 334: Table 3 space is missing. In table 3 there are also spaces missing between the genus and the species names

line 357: no s after hydrolyse.

line 367-8: please rewrite the sentence, which alcohol? what is meant with aromatics?

The supplementary also has a lot of typos, some chemical formulas are not in subscript, some words start with capitals while others not. Many species names lack a space between the genus and species names. It is Nocardia and not Nocaridia. It is Tween and not twain . Please describe what is meant with VB or give a reference. Just make everything look consistent. Table S2 might be better in landscape format to allow for more space for the species names

There are throughout the manuscript some weird commas. I hope the text editor will fix it.

The reference are not according to the style of the journal e.g in some cases issue number are given.

Please make sure the species names are correctly spelled throughout the manuscript.

Author Response

The link with the field of the Special Issue (Advance and Future Challenges to Microbial Food Safety) should be clearer. Consequently, could you revise, specially, the title and the abstract, to give the manuscript a proper/clearer orientation to the microbial food Safety of the baijiu, please?

Answer: Thank you for your kind advice. We have a request to replace the research topic (Advance and Future Challenges to Microbial Food Safety) with the research topic (New Insights in Microbial Diversity of Fermented Foods). We ensure that the title and abstract of the manuscript match the research topic.

For your guidance, itemized answers to each reviewer’s comments are appended below (answers are in blue font).

Answers to the Reviewer(s)' Comments:

1.line 28: please elaborate what is meant with aroma types

Answer: Thank you for your kind suggestion. According to the manufacturing techniques (traditional Baijiu, liquid fermentation Baijiu and traditional and liquid fermentation Baijiu), fermentation starters (daqu, xiaoqu, fuqu and mixed qu), and product flavors (more than 1870 volatile trace components such as esters, alcohols, acids, ketones and aldehydes), Baijiu products can be classified into different aroma types. The aroma types are directly led by different flavor compounds.

Corresponding references in the answers:

Zheng, X. W.; Han, B. Z. Baijiu (白酒), Chinese liquor: History, classification and manufacture. J.Ethnic Foods. 2016,3, 19-25.

Liu, H.; Sun, B. Effect of fermentation processing on the flavor of baijiu. J. Agric. Food. Chem. 2018,66 22, 5425-5432.

2.tables 1-2: inconsistent font, spacing issues, Brevibacterium typo, Shaanxi or Shanxi?, make sure names are widely used, e.g I could not find the names Nocardia soli and africana in literature. A lot of names are incorrectly spelled or there is no space between the genus and species name.

Answer: Thank you for your advice. We have modified the font formatting, table spacing, Brevibacterium spelling and province name (Shanxi). We re-checked the strain names of Nocardia soli and Nocardia africana and found that Nocardia africana exists but Nocardia soli is not present, Nocardia soli should be Nocardia xishanensis. We matched the species names and corrected spelling errors and spacing throughout the revised manuscript.

3.line 110: It is Brevibacteriumcelere

Answer: Thank you. We have replaced "Brevibacteriumceler" with "Brevibacteriumcelere " in the revised manuscript.

4.line 143: types of

Answer: Thank you for your comments. We have replaced "types" with "types of" in the revised manuscript.

5.table 2 at reference 64 it is colony and not colone

Answer: Thank you for your advice. According to re-examination of the reference, we have replaced "colone" with "colony " in table 2.

6.line 199-200: font issue with 10^7 and 1-^9

Answer: Thank you for your kind suggestion. We have made corresponding changes in the revised manuscript (lines 210-211).

7.line 201: better is not the best word to describe the diversity

Answer: Thank you. We have replaced "better than" with "superior to" in the revised manuscript in line 212

8.line 249: cells

Answer: Thank you for your advice. We have replaced "cell" with "cells" in the revised manuscript (line 260).

9.line 292: the growth of the latter

Answer: Thank you for your kind suggestion. We have made corresponding changes in the revised manuscript (line 303).

10.line 334: Table 3 space is missing. In table 3 there are also spaces missing between the genus and the species names

Answer: Thank you. We have filled in the missing spaces between the genus and the species names.

11.line 357: no s after hydrolyse.

Answer: Thank you. We have replaced "hydrolyses" with "hydrolyse" in the revised manuscript (line 369.

12.line 367-8: please rewrite the sentence, which alcohol? what is meant with aromatics?

Answer: Thank you. We have made the corresponding changes according to your kind suggestion. The three types of substances are listed in the revised manuscript (lines 380-383).

13.The supplementary also has a lot of typos, some chemical formulas are not in subscript, some words start with capitals while others not. Many species names lack a space between the genus and species names. It is Nocardia and not Nocaridia. It is Tween and not twain. Please describe what is meant with VB or give a reference. Just make everything look consistent. Table S2 might be better in landscape format to allow for more space for the species names

Answer: Thank you. According to your comments, we have modified the subscripts of the chemical formulas and the capitalization of some words, and added a space between the species name and the genus name and corrected the spelling of Nocaridia. We have replaced "twain" with "Tween" in the revised manuscript and add VB footnote to Table S1. We have changed the Table S2 from vertical to horizontal to have more space to show the strain names

14.There are throughout the manuscript some weird commas. I hope the text editor will fix it. The reference are not according to the style of the journal e.g in some cases issue number are given.

Answer: Thank you for your kind suggestion. We have modified the citation format of the references and have removed the issue number.

15.Please make sure the species names are correctly spelled throughout the manuscript.

Answer: Thank you very much for your suggestion, we matched each species name on the NCBI and corrected them one by one.

Round 2

Reviewer 1 Report

All the comments have been addressed by authors